# Interpretable (not just posthoc-explainable) medical claims modeling for discharge placement to reduce preventable all-cause readmissions or death

**Ted L. Chang[1,2], Hongjing Xia[1,2], Sonya Mahajan[1,2], Rohit Mahajan[1,2], Joe Maisog[3], Shashaank Vattikuti[4], Carson C. Chow[2,5], Joshua C. Chang[1,2,6]** *

**1** Sound Prediction Inc., Columbus, OH, United States of America, **2** Mederrata Research Inc., Columbus, OH, United States of America, **3** Lee Health, Fort Meyers, FL, United States of America, **4** Sleep Research Center, Walter Reed Army Institute of Research, Silver Spring, MD, United States of America, **5** Laboratory of Biological Modeling, NIDDK, National Institutes of Health, Bethesda, MD, United States of America, **6** Epidemiology and Biostatistics Section, Rehabilitation Medicine Department, The National Institutes of Health, Besthesda, MD, United States of America

* josh.chang@nih.gov

**Data Availability Statement:** This paper uses the CMS LDS, which the authors do not have permission to share. As per CMS: "The Centers for

## Abstract

We developed an inherently interpretable multilevel Bayesian framework for representing variation in regression coefficients that mimics the piecewise linearity of ReLU-activated deep neural networks. We used the framework to formulate a survival model for using medical claims to predict hospital readmission and death that focuses on discharge placement, adjusting for confounding in estimating causal local average treatment effects. We trained the model on a 5% sample of Medicare beneficiaries from 2008 and 2011, based on their 2009–2011 inpatient episodes (approximately 1.2 million), and then tested the model on 2012 episodes (approximately 400 thousand). The model scored an out-of-sample AUROC of approximately 0.75 on predicting all-cause readmissions—defined using official Centers for Medicare and Medicaid Services (CMS) methodology—or death within 30-days of discharge, being competitive against XGBoost and a Bayesian deep neural network, demonstrating that one need-not sacrifice interpretability for accuracy. Crucially, as a regression model, it provides what blackboxes cannot—its exact gold-standard global interpretation, explicitly defining how the model performs its internal "reasoning" for mapping the input data features to predictions. In doing so, we identify relative risk factors and quantify the effect of discharge placement. We also show that the posthoc explainer SHAP provides explanations that are inconsistent with the ground truth model reasoning that our model readily admits.

## Introduction

Preventable readmission after hospital discharge is costly. In 2011, for adult 30-day all cause hospital readmission in the United States, the cost was about $41.3 billion [1]. To improve

Medicare & Medicaid Services (CMS) makes Limited Data Set (LDS) files available to researchers as allowed by federal laws and regulations as well as CMS policy. LDS files contain beneficiary-level health information and are considered identifiable files, but they do not contain specific direct identifiers as defined in the Health Insurance Portability and Accountability Act (HIPAA) Privacy Rule. Questions about LDS files or the process for requesting LDS files can be sent to datauseagreement@cms.hhs.gov.

**Funding:** CCC is supported by the Intramural Research Program of the NIH, NIDDK. JCC is partially supported by the Intramural Research Program of the NIH, Clinical Center. This work used the Extreme Science and Engineering Discovery Environment (XSEDE) [58], which is supported by National Science Foundation grant number ACI-1548562 through allocation TG-DMS190042.

**Competing interests:** The authors have declared that no competing interests exist.

outcomes, Medicare, through its Hospital Readmissions Reduction Program [2], penalizes providers for readmissions that occur within the 30-days after discharge; penalties have spurred interest in interventions surrounding transitions of care including discharge planning services such as hand-offs to less-intensive healthcare institutions. Population-scale individual-level medical claims data provides rich longitudinal health context behind each hospital stay, making it possible to assess the efficacy of these interventions retroactively. This manuscript focuses on the problem of deciding discharge placement for individuals in order to prevent readmission or death.

## Readmission models

A recent review [3] surveyed properties of readmission models in the literature. By and large, they found no model type to consistently predict more-accurately than others. Some studies have reported marginal improvements using either XGBoost or neural networks over interpretable methods [4–6], though not consistently [7–11], as seen in other problems [12]. Generally, the literature has focused on 30-day readmissions, though nuances in how readmission is defined complicate direct performance comparisons. Models based on medical claims data typically achieved area under the receiver operator characteristic (AUROC) of approximately 0.7 for predicting their version of all cause 30-day readmission.

Another factor that complicates the direct comparison of modeling efforts is differences in datasets—and hence the underlying patient populations and predictors. We are aware of two readmission studies performed on datasets identical to ours. MacKay et al. [13] developed XGBoost models for predicting a set of adverse events, reporting an AUROC of 0.73 for all-cause readmission prediction. Lahlou et al. [14] created an attention-based neural network for predicting admissions after discharge within 30-days and reported an AUROC value of 0.81, however, they did not distinguish between transfers, planned admissions, and acute admissions in their outcome label so they solve a different problem that is of less practical utility.

Yet, having a high AUROC is insufficient for making a model useful. Prerequisites for utility include the ability to understand predictions, assess validity, and derive actions. Model interpretability is a means to these ends. Most studies surveyed were aware of the importance of model interpretability, regardless of whether they produced interpretable models. Studies that claim interpretability for their blackbox solutions only offer "posthoc explainability," a catch-all phrase for narratives generated in order to promote a sense that a model is interpretable when it is not.

## Blackbox models

Methods such as Deep Learning (DL) and ensemble boosted trees (XGBoost, LightGBM, others) can model nonlinearities. When copious training data is available, these methods yield models that are more expressive than traditional generalized linear models. Most-generally, blackbox models like DL and ensemble trees are nonlinear kernel machines (function interpolations) [15]. The convoluted nature of their interpolations makes these models uninterpretable. Massive investment exists in these models because of their predictive performance and low effort requirement. This existing investment, the challenge of creating truly interpretable models, and a myth that blackboxes perform better than interpretable models [16], incentivize the marketing of posthoc-xAI as an alternative to interpretable modeling. In finance, a similarly high-stakes domain, there has been wide resistance to blackbox modeling, formalized recently in model risk management guidelines published by The Office of the Comptroller of the Currency (OCC) [17]. We should also be wary of the use of these models in healthcare, where the risk to patients requires truly trustworthy solutions.

## Interpretability

The goal of interpretable modeling is to produce predictions that an end-user can understand [16, 18], which is a prerequisite for making a prediction actionable. One necessary yet insufficient aspect of intrinsic model interpretability is feature attribution. Blackbox models do not admit feature attributions without the use of unreliable approximations. Conversely, feature attribution is exact in regression models, where each model coefficient has the unequivocal interpretation as the conditional expected change of the response corresponding to a given unit of change in the predictor, while fixing the other predictors. For this reason, even ignoring attributes beyond feature attribution, a significant disconnect already separates blackbox models from inherently interpretable models. Fig 1 is a representation of the spectrum of interpretability focusing on structured data problems in healthcare.

While the definition of interpretability varies according to problem domain, all notions of interpretability require a basic ability to parse the computations behind a model's predictions in terms of the input data features. We refer to this fundamental aspect of interpretability as "computational interpretability." Computational interpretability is a necessary yet insufficient attribute for prediction comprehensibility. ReLU-activated neural networks, matrix composition methods like principle components analysis (PCA), and large multiple regression models are computationally interpretable, whereas Deep Learning (DL) models in-general and ensemble trees methods like XGBoost are not.

However, knowing how a prediction is computed from individual features does not automatically make the prediction comprehensible—it may still difficult to understand how a model behaves as there is a limit to the capacity of information that humans can process simultaneously [19]. Sudjianto et al. [20] note that additivity, sparsity, linearity, smoothness, monotonicity, and visualizability are some attributes of interpretable models that are also comprehensible. Each of these attributes can be enforced through suitable modeling constraints.

The highest bar for interpretability is for a model to be mechanistically meaningful. These models often leverage domain knowledge and are capable of providing deep and robust insights. They also often justify causal interpretations [21]. Even if one can truly understand a model, one often cannot act on it. To be directly actionable, a model also needs to adjust for biases in the data so that its prediction of the effects of interventions can be interpreted causally [22]. Yet, independent of causal validity, predictive model interpretability is still important because it allows practitioners to better understand the risks and biases of a given model.

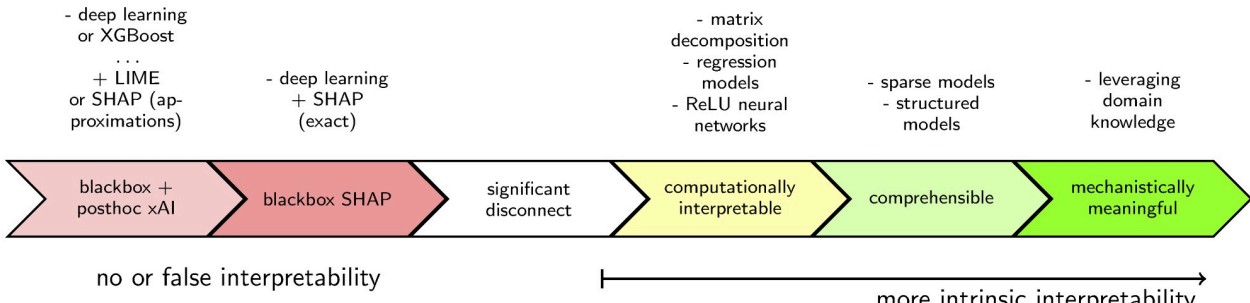

**Fig 1. Model interpretability lies along a spectrum with a clear chasm existing between intrinsically interpretable models others.** Models without intrinsic interpretability rely on unreliable approximation techniques for crafting explanations. More-interpretable models are more trustworthy and insightful.

## Posthoc explainable-AI (xAI)

Posthoc xAI is a set of techniques to market uninterpretable blackbox models as interpretable (Fig 1). The most popular xAI methods (LIME [23] and SHAP [24, 25]), use approximations [26, 27] to provide narratives of feature importance within a prediction. Other methods such as attention [28] build an explanation mechanism as a module within a blackbox model in order to more-easily compute them [29, 30]. Narratives, convincing as they might seem, are not necessarily true. In fact, researchers have shown [30–35] that these methods provide imprecise and unreliable explanations of models, and often disagree. Aptly, Krishna et al. [36] coined this the "disagreement problem" with posthoc interpretability and conducted a survey of real world data scientists finding no consistent or principled method to handle these inconsistencies. As Rudin [16] notes, "an explanation model that is correct 90% of the time is wrong 10% of the time." Despite marketing claims, xAI does not carry blackbox models across even a very minimal bar of requirements for interpretability. If an explanation is not true to one's model, any sense that the model is comprehensible is based on faulty information.

## Piecewise-linear modeling

Blackboxes provide clues on how to extend traditional linear models. DL is the application of artificial neural networks (ANNs) to prediction problems. ANNs consist of sequences (or more generally of graphs) of successive affine matrix arithmetic operations, sandwiched between activation functions. In general, these methods are blackboxes, with the exception of ReLU-activated neural networks (ReLU-nets for short). Examining ReLU-nets elucidates the nature of how DL captures nonlinearities. ReLU-nets use the activation function

$$\text{ReLU}(x) = \ 0 \ \text{if} \ x \leq 0 \quad \text{or} \quad x \ \text{otherwise.} \tag{1}$$

In these models, ReLU is independently applied to each matrix coordinate after each successive matrix operation. The output of the function is nonzero if and only if a linear combination of the elements computed by the prior layer are positive. Hence, ReLU defines an inequality over quantities within the model—applied to each coordinate within each layer, ReLU defines recursive sets of inequalities. These inequalities collectively segment the training data into disjoint regions. In sum, ReLU-nets are composed of regionally-disjoint generalized linear models—each of which is interpreted in the same manner as linear regression. Hence, ReLU-nets are computationally interpretable. The salient nonlinearity of these models is locality. To interpret a specific prediction given by these models, one needs to map the input to a particular linear submodel. Then, conditional on this mapping, a ReLU net is locally a simple generalized multiple linear regression model. Observing this fact, Sudjianto et al. [37] provides a tool for exactly interpreting trained ReLU neural networks, by unwrapping the cascades of inequalities. In this manuscript we mimic this property of ReLU-nets within a well-controlled multilevel Bayesian regression framework in order to gain expressiveness while prioritizing interpretability.

## Methods

We generalize the classic readmission problem of within 30 days of discharge, to the likelihood of readmission at any arbitrary day after discharge. To this end, our objective is to characterize the statistics of the inter-inpatient wait time $T_n$. Additionally, we focus on identifying the effects of discharge placement, representing the choices symbolically as $I_n$, ranked in terms of health acuity: (0) discharge home, (1) discharge home with home health, (2) discharge to skilled nursing, (3) intermediate care/critical access, (4) long term care, (5) other less-acute

inpatient. The issue that complicates the estimation of discharge placement effects is unobserved confounding—providers use the patient's health status in order to decide placement. To resolve the treatment assignment bias, we model the joint outcomes

$$T_n \sim f(T_n | \boldsymbol{x}_n, \boldsymbol{I}_n, \boldsymbol{\alpha}_n, \boldsymbol{\beta}_n, \boldsymbol{\gamma}_n)$$
$$I_n \sim g(I_n | \boldsymbol{x}_n, \boldsymbol{v}_n, \boldsymbol{\xi}_n),$$

(2)

where $\boldsymbol{x}_n \in \mathbb{R}^p$ is a covariate vector and we explicitly adjust for assignment bias. Note that we distinguish between the scalar-valued $I_n$, which corresponds to the list of interventions above, and the vector valued $\boldsymbol{I}_n$ which we will explain later in this manuscript. For the sake of interpretability, we formulate $f$ and $g$ in Eq 2 as hierarchical multilevel Bayesian generalized linear regression models, However, to increase expressivity by introducing the type of nonlinearity seen in ReLU-nets, we allow all of the model parameters $\boldsymbol{\alpha}_n, \boldsymbol{\beta}_n, \boldsymbol{\gamma}_n, \boldsymbol{v}_n, \boldsymbol{\xi}_n$ to vary locally [38–40] across regions defined by $x_n$, in ways that comport with domain knowledge.

## Ethics statement

This study used the de-identified Centers for Medicare and Medicaid Services (CMS) Limited Dataset and was commissioned by the CMS for their inaugural AI competition. As a result, it is not subject to IRB approval. The CMS provided data for this study at the commencement of the competition in December 2019, and again in June 2020 for model evaluation. The provided dataset encompassed a 5% sample of adult Medicare beneficiaries in the USA.

## Data preprocessing

The available dataset, the CMS Limited Dataset (CMS LDS), consists of a national 5% beneficiary sample of Medicare FFS Part A and B claims from 2008 to 2012. The 2008 claims had only quarter date specificity so we used them solely to fill out the medical history for 2009 inpatient stays, by assuming that each 2008 claim fell in the middle of its given quarter. We trained the readmission models on 2009—2011 admissions, and evaluated the models on 2012 admissions.

After grouping claims into coherent episodes, based on date, provider, and patient overlap, we filtered for inpatient-specific episodes with certain characteristics to use as index admissions. We retained only episodes where the patient had a continuous prior year of Part A/B enrollment. We also excluded episodes from consideration as index episodes if they did not correspond to discharges to less-intensive care (excluding death and most inpatient-to-inpatient transfers). Additionally, we used the official CMS methodology for determining whether each episode is a planned admission, acute admission, or potentially planned admission [41]. For each episode we then computed the waiting time to either the next unplanned acute episode or death, or until censorship due to the end of the observation window. In the end, the training dataset consisted of approximately 1.2 million inpatient episodes, of which approximately 17% were followed by an unplanned acute inpatient episode and an additional 3% by death, within 30 days.

For each episode, we collected all billing codes, creating lists of concurrent procedure and diagnostic codes. Additionally, we collected the preceding four quarters of history for each episode, aggregating billing codes on a lagged quarterly basis.

**Feature engineering.** Medical claims data consists of series of billing codes in several dialects (ICD9/10, HCPCS, RUG, HIPPS, etc). We down-sampled diagnostic (Dx) and procedure (Tx) codes, from their original dialects to multilevel Clinical Classification Software (CCS) codes [42]. CCS codes are clinically curated hierarchical categories that are more tractable for analysis and interpretation. Mapping to CCS drastically removes redundancy in the

vocabulary of the dataset and helps to separate the health-specific information in billing codes from noisy reimbursement-specific details.

We used AHRQ Healthcare Cost and Utilization Project (HCUP) databases in order to tag codes for comorbidities, chronic conditions, surgical flags, utilization flags, and procedure flags. Included within skilled nursing facility (SNF) and home health (HH) claim codes are also activities of daily living (ADL) assessments. We converted these codes to ADL scores, where higher scores correspond to lower functional ability. We also incorporated CMS's risk adjustment methodology, hierarchical condition categories (HCC), as model predictors. The CMS LDS contains beneficiary county codes that we used to incorporate the urban rural index and social economic scale as model features. Together with beneficiary race information and Medicaid state buy-in, these variables allowed for some measure of social determinants of health.

We encoded CCS and other code mappings into numerical vectors by counting the incidences of each permissible code. In the case of CCS, which is multilevel, we truncated codes at each of the first two levels and counted at each level. Altogether, the numerically encoded derived features constituted a vector of size $p = 1072$, which encompassed both concurrent episode codes and the past four quarters of history, where CCS was truncated to the first level for history.

**Feature quantization.**   To improve model interpretability, we made an effort to place all model parameters (log hazard ratios) on the same scale so that the magnitudes of all regression coefficients are directly comparable. In examining our derived data features, we found that they were predominantly sparse and heavy tailed. When fitting a logistic regression model to these data features, the model fit poorly to observations with large counts. Theses findings, and our desire to optimize model interpretability, led us to quantize all numerical variables so that the input variables into the model are entirely binary. To this end, we first computed the percentiles for each feature across the entire dataset. Then we re-coded each quantity into a series of binary variables corresponding to inequalities, where the cutoffs were determined by examining each variables at a set of quantiles and eliminating duplicate values. The usage of quantile-based coding has appeared in the literature [43, 44] as a nonlinear feature coding that has demonstrated benefits to model performance in certain problems. Generally, we retained only the quantized features in specifying the models except when otherwise specified. The total size of the feature vector after dropping all original non-quantized numerical features and all constant features expanded to $p = 3143$.

## Survival modeling

For flexibly modeling the wait time distribution $f$, we use the piecewise exponential survival regression model (PEM) [45]. PEMs are defined by specifying the time-dependent hazard $\lambda(t) : \mathbb{R}^+ \rightarrow \mathbb{R}^+$ using a piecewise constant function, where the hazard changes across breakpoints that define disjoint time intervals. The probability density function for the PEM follows

$$f(t) = \lambda(t)e^{-\int_0^t \lambda(u)\mathrm{d}u}. \tag{3}$$

In this manuscript we set the breakpoints between time intervals at 1 week, 4 weeks, and 9 weeks after discharge. For each episode $n$, we can estimate a wait time distribution by estimating the log-hazard within each time interval $i$,

$$\log \lambda_{ni} = \alpha_{ni} + \boldsymbol{\beta}'_{ni}\boldsymbol{x}_n + \boldsymbol{\gamma}'_{ni}\boldsymbol{I}_n, \tag{4}$$

where we allow the model parameters to vary across the data regionally, in a manner that emulates the type of nonlinearity seen in ReLU-nets. In Eq 4 we separate out the discharge

placement effects ($\gamma_n$) from other effects ($\beta_n$). Doing so makes it easier to structure the model for causally interpreting the discharge assignment effects. We incorporate domain knowledge by acuity-ordering the interventions, enforcing monotonicity of intervention effect by constraining the last five coefficients of $\xi_n$ to non-positivity.

We model the discharge placement process $g$ using an ordinal logistic regression model, where

$$
\begin{aligned}
I_n | \boldsymbol{p}_n &\sim \text{Categorical}(p_{n0}, \ldots, p_{n5}) \\
p_{nk} | \boldsymbol{x}_n, \boldsymbol{v}_n, \boldsymbol{\xi}_n &= \Pr(I_n \geq k | \boldsymbol{x}_n, \boldsymbol{v}_n, \boldsymbol{\xi}_n) \\
&\quad - \Pr(I_n \geq k+1 | \boldsymbol{x}_n, \boldsymbol{v}_n, \boldsymbol{\xi}_n) \\
\Pr(I_n \geq k | \boldsymbol{x}_n, \boldsymbol{v}_n, \boldsymbol{\xi}_n) &= \text{logit}^{-1}(v_{nk} + \boldsymbol{\xi}'_n \boldsymbol{x}_n),
\end{aligned}
\tag{5}
$$

where $\boldsymbol{\xi}_n$ are slopes corresponding to episode $n$ and $\boldsymbol{v}_n = [v_{n1}, \ldots, v_{n5}]$ are intercepts under the constraints $v_{nk} < v_{n,k+1}, \forall k, n$. The predictions given by this model then feed back into the prediction of the wait time through a slope term for each element of the covariate vector $\boldsymbol{I}_n = [\Pr(I_n \geq 1 | \ldots), \ldots, \Pr(I_n \geq 5 | \ldots), 1_{I_n \geq 1}, \ldots, 1_{I_n \geq 5}]$. Utilizing the discharge placement probabilities as model covariates adjusts for the confounding bias caused by the selection process, in a manner analogous to incorporating the local treatment probability as a covariate [46]. Additionally, directly modeling the treatment effects within a multilevel model allows us to infer locally-varying treatment effects, partially pooled for stable inference in regions where the data is sparse [47, 48].

**Parameter decomposition.** The piecewise linear nature of ReLU-nets, and the observation that neural networks produce learned data representations [49], suggest that an approach to mimicking their expressivity within regression models is to allows slopes (and intercepts) to vary across regions of the data. We do so by expressing each of these regionally-varying parameters using an additive decomposition.

First, for delineating regions in data space (corresponding to cohorts), we project portions of the input data to lower dimensions through unsupervised methods. In Chang et al. [50], the authors make a connection between sparse probabilistic matrix factorization and probabilistic autoencoders. We use this approach to develop a low-dimensional representation of the portions of the input covariate vector that pertain to the lagged quarterly history. Then, we compute the statistics of the learned representation in the training data and develop for each dimension a set of cut-offs to use for bucketization. This procedure puts each inpatient episode into a specific cohort, represented by a location within a multidimensional lattice, based on medical history. Specifically, we used a single cut-off (the median) for each of five dimensions (S2 Fig in S1 File), creating a set of $2^5 = 32$ groups based on history. By design, the rules governing the group assignment can be easily converted to a set of inequalities over sparse subsets of the original data features. Additionally, we included interactions between the history groups with other discrete attributes such as the major diagnostic category (MDC), complication or comorbidity (CC) or a major complication or comorbidity (MCC), and race, to create high dimensional discrete lattices where the cells define coarse interaction cohorts in the data. When partitioning data by a high-order interaction, a big data problem quickly becomes many small data problems—divide-and-conquer approaches can suffer from overfitting. To combat this issue, we developed a multiscale modeling approach where higher-order interactions are regularized by partially pooling their effects into related lower-order interactions. Specifically, given a multidimensional lattice that represents all cohorts for which the parameter will vary,

we assign for each parameter a value within the lattice by decomposing the value into the form

$$
\theta^{(\boldsymbol{\kappa})} = \overbrace{\theta^{(*,*,\ldots,*)}}^{\text{zero order}} + \overbrace{\theta^{(\kappa_1,*,\ldots,*)} + \theta^{(*,\kappa_2,*,\ldots,*)} + \ldots}^{\text{first order}}
$$
$$
+ \overbrace{\theta^{(\kappa_1,\kappa_2,\ldots,*)} + \theta^{(\kappa_1,*,\kappa_3,*,\ldots,*)} + \ldots}^{\text{second order}} + \text{H.O.T.},
$$

(6)

where $\boldsymbol{\kappa} = (\kappa_1, \kappa_2, \ldots, \kappa_D)$ is a $D$ dimensional multi-index. In practice, we truncate the maximum order of terms in this decomposition due to memory constraints. More details on the exact decompositions that we used for our model parameters can be found in the Supplemental Materials.

**Regularization.** By design, the parameter decomposition method inherently regularizes by partial pooling [47]. Additionally, we used weakly informative priors on the component tensors in these decompositions in order to encourage shrinkage at higher orders. For the regression coefficients, we utilized the horseshoe prior for local-global shrinkage [51–54]. Please see the Supplemental Materials for more details on the model specification.

**Relationship to other interpretable model types.** In the Introduction, we motivated our methodology by noting the piecewise-linear nature of ReLU activated artificial neural networks. Those models offer a particular type of local linearity that we intended to mimic—the main improvements of our methodology are in moving beyond local linear interpretability by making the mapping for a data point to a local region easier to understand, and in explicitly exploiting the partial pooling properties of hierarchical mixed effects models.

Beyond ReLU-nets, other inherently interpretable models, where nonlinearity results from the locality of relationships, exist under the wide umbrella of varying coefficient regression models [39, 55] including hierarchical mixed effects models, tree-boosted varying coefficient regresion models [56], and a broad class of ensemble-like models that can be formed by local procedures such as hierarchical stacking [57]. Additionally, inherently interpretable globally nonlinear models are also popular—many of these models are extensions to classical generalized additive models (GAMs) [58], including explainable boosting machines [59] and the ReLU-net powered GAMI-Net [60].

Each of these methodologies offer computational interpretability (Fig 1). Through suitable constraints (human intervention) many of them can be tuned so that the resulting "reasoning" that a model is performing is comprehensible and/or mechanistically meaningful.

**Implementation.** We used Tensorflow Probability [61], developing a set of libraries for managing the parameter decompositions that is publicly available at `github:mederrata/bayesianquilts`.

We trained our model using minibatch mean-field stochastic ADVI, using batch sizes of $10^4$, and a parameter sample size of 8 for approximating the variational loss function. We utilized the Adam optimizer with a starting learning rate of 0.0015, embedded within a lookahead optimizer [62] for stability. Each epoch where the mean batch loss did not decrease, we set the learning rate to decay by 10%. Training was set to conclude if there was no improvement for 5 epochs, or if we reached 100 epochs, whichever came sooner. More information on the training is present in the Supplemental Materials. We used scikit-learn 1.1.1 for fitting two baseline logistic regression models (all features and restrcited to only LACE features), and XGBoost 1.6.1 for fitting a reference blackbox model for comparison. We implemented a horseshoe Bayesian convolution neural network with ReLU activation using TFP, where we used a single hidden layer of size one-fifth the input layer. For computing global SHAP values, we used regression-based KernelSHAP [63]. All computation was performed using the Pittsburgh

Supercomputing Center's Bridges2 resources. We utilized extreme memory (EM) nodes for preprocessing, and Bridges2-GPU-AI for training.

## Results

### Prediction accuracy

Table 1 shows the classification accuracy of our model in predicting readmissions or death within the first 30 days, benchmarked against predictions given by alternative models trained on the same dataset. Note that the LACE model is also trained using our dataset, restricted to LACE predictors [64]. The standard deviation in both the AUROC and AUPRC measures, as determined using bootstrap, was approximately 0.003. Non-linearly transforming our count features using quantization improved the accuracy of logistic regression to nearly match that of XGBoost on this dataset as measured by AUROC. Hence, we used quantization for features in both the Bayesian neural network (BNN) and piecewise exponential (PEM) models. The Bayesian neural network we developed utilizes sparsity-inducing horseshoe priors [65] on the weights and biases, which has been shown to improve model performance [52].

### Interpretation

In addition to being competitive with blackbox methods in terms of prediction accuracy, our model, as a generalized linear survival regression model, is easily interpretable. To be specific, our model is a generalized linear survival model where the coefficients vary. The value of each coefficient is the logarithm of a hazard ratio corresponding to the effect of a given feature, for a given data cohort, for a given time period. Log hazards greater than zero correspond to increased probability of event (readmission or death). Here, we provide select portions of the *ground-truth* global interpretation of the model, found by simply reading off the values of the regression coefficients. Please see the Supplemental Materials for a more-complete accounting

**Table 1. 30-day unplanned readmission or death classification metrics for evaluated models: XGBoost, Sparse logistic regression (LR), Bayesian neural network (BNN), our Piecewise exponential model (PEM).** Quantization refers to the histogram-based bucketization of real-valued features. Area under the receiver operator curve (AUROC) and area under the precision-recall curve (AUPRC) computed on held-out 2012 inpatient episodes. Models trained on 2009–2011 episodes. Interpretability judged according to Fig 1.

| Model | Interpretability | AUROC/AUPRC | |
|---|---|---|---|
| **Classification models** | | 30-day | |
| XGBoost w/o quantization | None | 0.741 / 0.465 | |
| ReLU-BNN classifier | Computationally | 0.750 / 0.481 | |
| LR classifier—LACE only | Comprehensibly | 0.666 / 0.313 | |
| LR classifier w/o quantization | Comprehensibly | 0.666 / 0.484 | |
| LR classifier | Comprehensibly | 0.747 / 0.448 | |
| **Survival models** | | 30-day | 90-day |
| Exponential ReLU-BNN | Computationally | 0.730 / 0.410 | 0.753 / 0.612 |
| Exponential Cox PH | Comprehensibly | 0.729 / 0.403 | 0.753 / 0.606 |
| Exponential Quilt | Comprehensibly | 0.744 / 0.468 | 0.760 / 0.632 |
| Weibull AFT | Comprehensibly | 0.533 / 0.207 | 0.530 / 0.362 |
| Weibull ReLU-BNN | Comprehensibly | 0.649 / 0.296 | 0.676 / 0.503 |
| Weibull Quilt | Comprehensibly | 0.688 / 0.334 | 0.717 / 0.551 |
| Generalized Gamma Quilt | Comprehensibly | 0.704 / 0.345 | 0.718 / 0.554 |
| PEM quilt prediction-only | Comprehensibly | 0.751 / 0.477 | 0.765 / 0.638 |
| **PEM quilt (discharge)** | Mechanistically | 0.750 / 0.475 | 0.764 / 0.638 |

of the model. This type of exposition is impossible with blackbox models without relying on unreliable approximations.

**Time-dependent risk factors.** The model segments the data based on low-dimensional representation and assigns for each predictor a cohort-level effect within each time interval. The cohorts are delineated by the recent history of medical services utilization and the properties of the present hospital admission. The effects within the model are hazard ratios, which describe the instantaneous relative risk associated with a predictor relative to a baseline. In most cases, the baseline refers to a typical or normal value of a variable. Membership to cohorts also itself is associated with a baseline risk—baseline log-hazards are presented in Fig 2 for the 12480 episode cohort types defined within the decomposition for the parameter vector $\boldsymbol{\alpha}_n$ in Eq 4. Larger values of the hazard imply higher probability of event (readmission or death). There exists variability in the hazards across cohorts (rows), though the most striking change is in time (columns). Generally, the hazard is greatest in the first week after discharge. This finding implies that patients are more vulnerable in the first week than afterwards—keeping a patient out of the hospital within the first week has the largest impact on the overall risk that they will die or be readmitted. For this reason, we will focus on understanding the model's predictions of the first-week risk.

The 40 most-impactful first-week factors are shown in Fig 3, where the parameters have been decomposed in order to control for racial biases. The most-predictive single feature was length of stay. Lengths of stay less than a full day had a relative log hazard ratio of 0.97 (95% CI: 0.96–0.98) (note LOS<1 day was the reference group and so is the converse of LOS $\geq$ 1 days shown in Fig 3). Having an acute primary diagnosis code, at least one inpatient stay in the

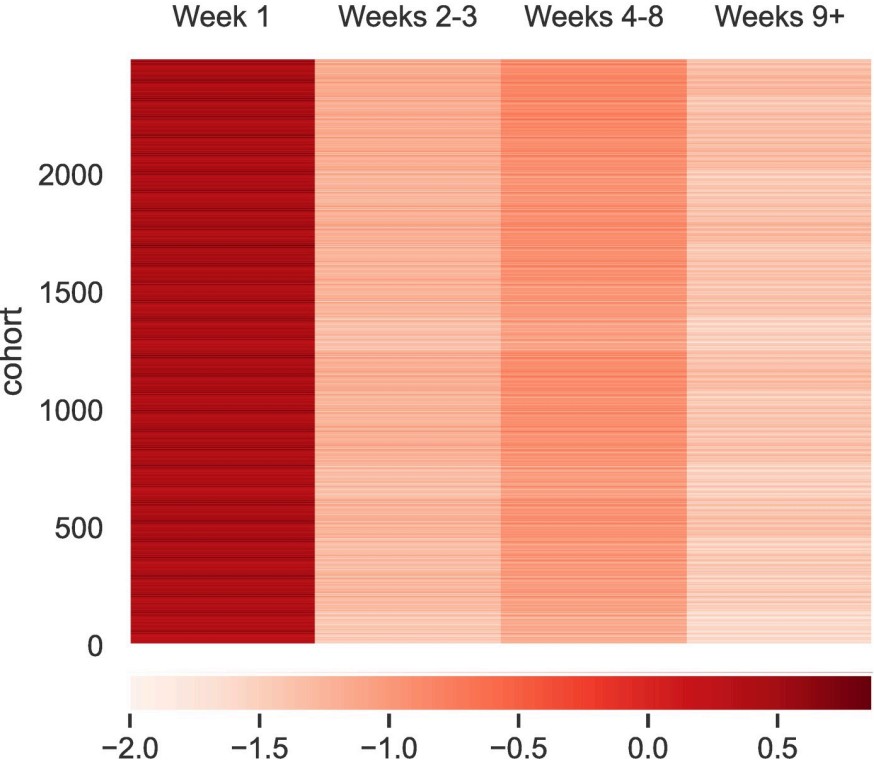

**Fig 2. Mean baseline log-hazards by week for each episode interaction cohort defined within the model.** Larger log-hazards corresponds to more readmission risk. Personalized values of $\alpha_n$ specific to each episode are found by mapping an episode into its cohort grouping.

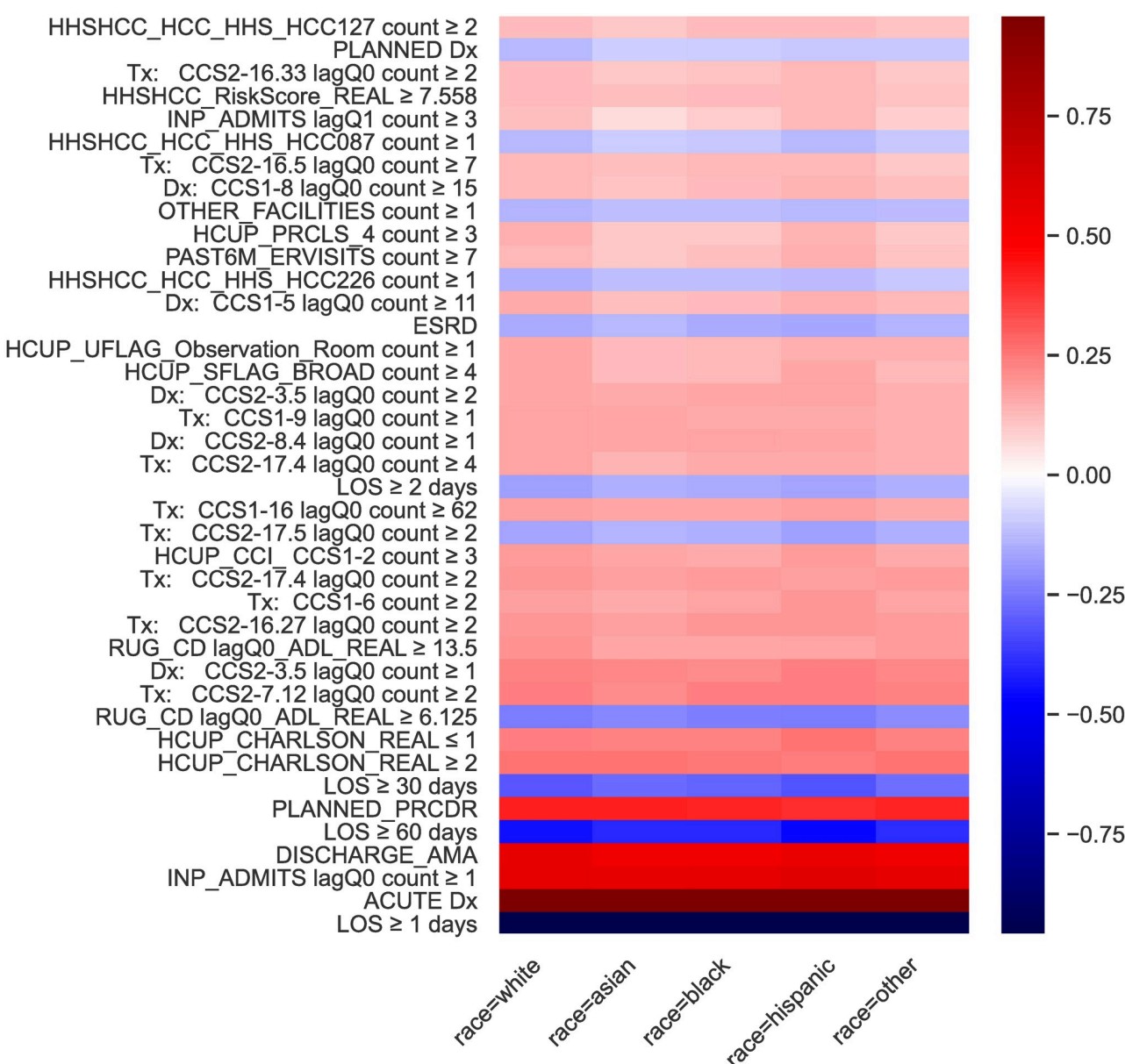

**Fig 3. The 40 predictors with the largest absolute coefficients in the first week (through day 7) after readmission.** All predictors are binary and all parameters are additive log hazard ratios. Higher (red) corresponds to larger hazards and greater readmission risk.

previous quarter (lagQ0, within 90 days of admit), and discharge against medical advice were also strong predictors associated with increased risk of readmission or death. Patients who received skilled nursing care in the quarter preceding an inpatient episode, who had a Resource Utilization Group (RUG) Activities of Daily Living (ADL) score of at least 6.125 tended to have a lower risk of readmission in the first week than otherwise, however, the risk increased for quarter-lagged ADL scores of at least 13.5.

**Discharge placement effects.** In Fig 4, we show the cohort-wise causally-adjusted mean local average treatment effects of discharge to each of the given care settings as well as the local standard deviation in the effect. Focusing on the effect of discharging to skilled nursing care,

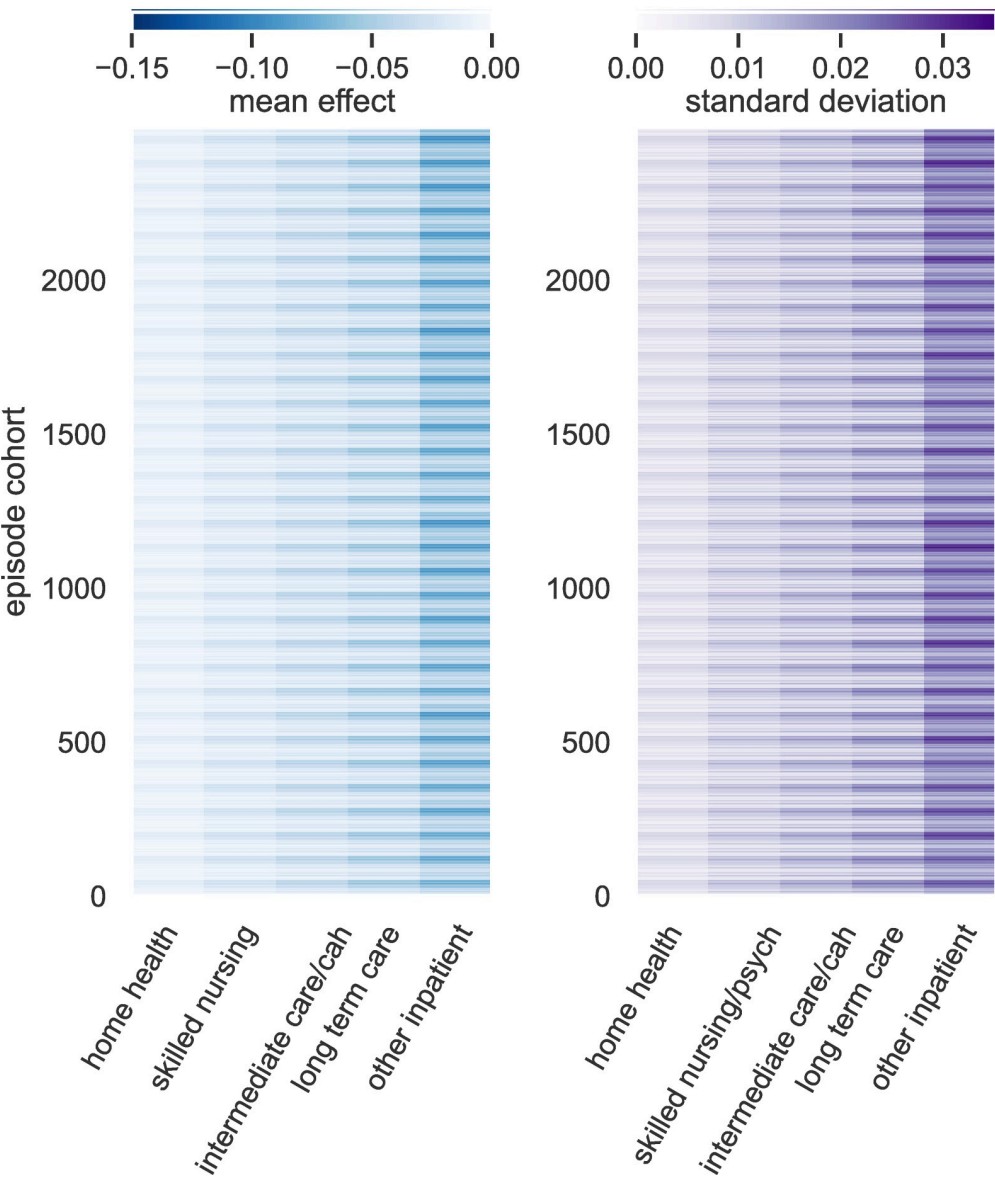

**Fig 4.** First-week effects of discharge placement: Mean (left) and standard deviation (right) by cohort (row) of the five placement interventions assessed, in increasing order of implied acuity. Effect is difference in log-hazard relative to a normal discharge (home).

the effects were greatest for episodes graded by DRG code as having either a complication or comorbidity (CC) or a major complication or comorbidity (MCC). In particular, CC/MCC episodes with a major diagnostic code of 2 (Diseases and Disorders of the Eye), 14 (Pregnancy, Childbirth And Puerperium), and 22 (Burns) have the greatest response to discharge to skilled nursing.

## Posthoc-xAI (SHAP) misleads

Knowing what the model is doing in exact terms, let us see how posthoc-xAI thinks the model is working. In Fig 5 we display the most important model features as determined by magnitude

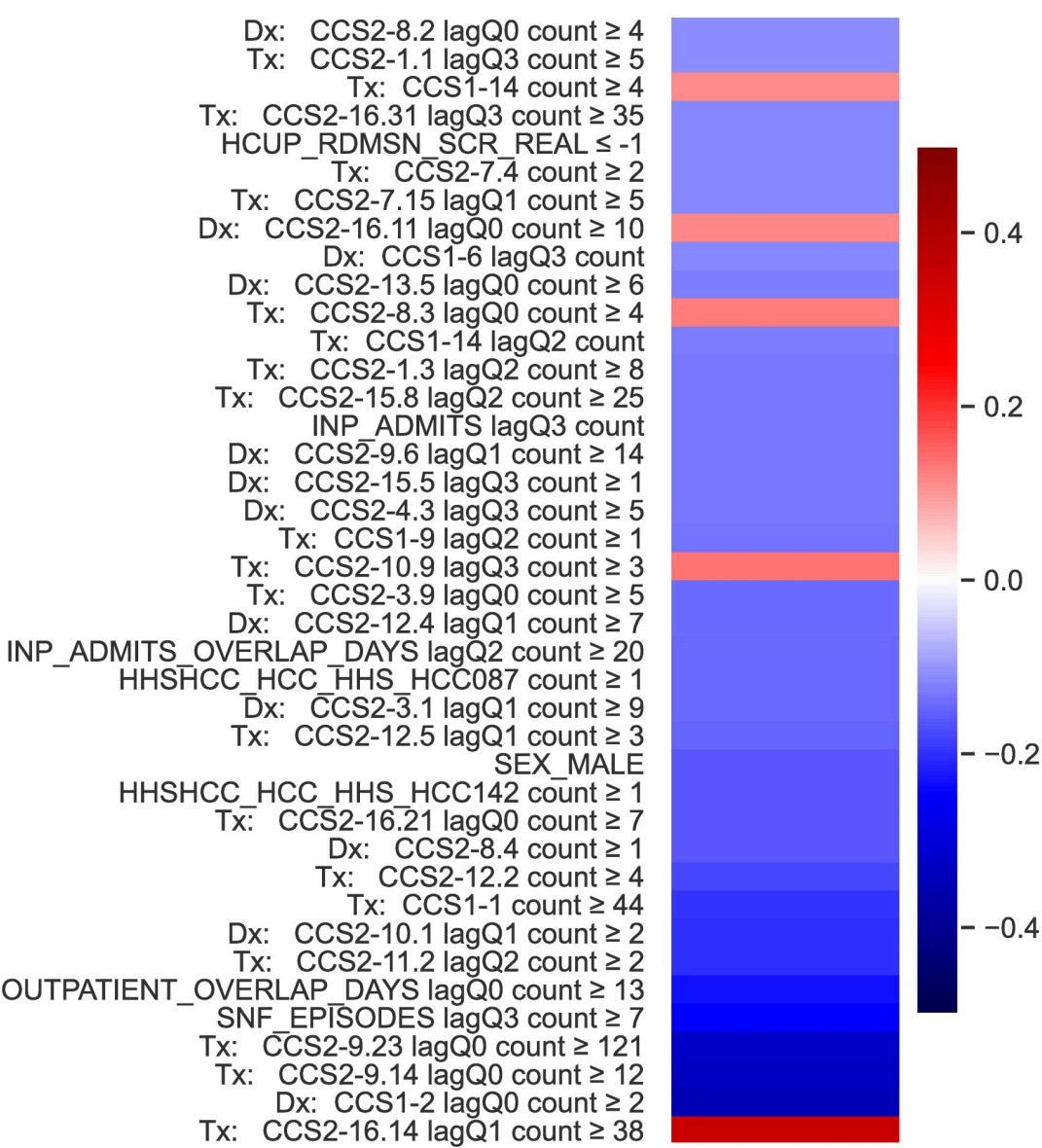

**Fig 5. Shapley values: Top 40 absolute feature weights for prediction of 30 day readmission using our survival model, where we have the ground truth explanation.** See Fig 4 and S2 Fig in S1 File, and the supplement for how the features actually are incorporated into the model. SHAP fails to identify the features a model is using whenever features are correlated.

of global SHAP values in the prediction of readmission or death within the first 30 days. SHAP is computationally costly to approximate—the details of our SHAP computation are available in the Supplemental Materials. The four most-influential features according to the explainer are specific CCS classes of treatments and diagnoses in the recent quarterly history. Comparing these results to the parameter values of Fig 3, it is evident that the feature sets disagree. Nor do the values in Fig 5 align with parameter values for later weeks (see Supplemental Materials). This finding is unsurprising; SHAP has been consistently shown to fail to recover ground-truth interpretations [32, 66], in problems where the predictors are correlated. SHAP fundamentally does not answer the question of what a given model is doing in order to reach a

prediction. Furthermore, feature importance is not grounded in any relevant units and also does not speak to relevant interactions that are captured in a model. We criticize SHAP because it is one of the most popular posthoc-xAI techniques, however, similar arguments hold for other techniques [16, 30, 67].

## Discussion

We presented a method for mimicking ReLU-nets within inherently interpretable multilevel Bayesian models. We applied this methodology to the prediction of hospital readmissions or death after discharge, and to the causal inference of the effects of discharge assignments.

### Accuracy without blackboxes

We demonstrated how we were able to perform like blackboxes, without sacrificing interpretability. We accomplished this feat through two classes of methods: First, our novel modeling framework allowed us some fine-grain resolution in looking at the differential effects of the predictors in data subgroups. Additionally, it helped regularize the inference of local average treatment effects for choosing discharge placement. Second, we performed layers of feature engineering. The first layer was an extraction of medically-relevant information from the raw billing that gave us attributes such as chronic diseases, comorbidities, and ADL function. Then, we reduced noise in the raw coding by mapping to the clinically-relevant CCS system. These two steps were sufficient for our logistic regression model to match the performance of an XGBoost model in the literature based on the same dataset [13]. Finally, we performed feature quantization based on the per-feature statistics. Quantization led to a big performance increase in logistic regression and also in the neural network for a given model size. We took these lessons and used them in defining our interpretable survival model.

### Posthoc xAI is inherently untrustworthy

Our model, being a regression model is inherently interpretable. It admits an unequivocal ground-truth explanation that is found by simply examining its regression coefficients, all of which are log hazard ratios. Hence, it is a good test case for testing the accuracy of posthoc explainers. We tested SHAP on our model; it failed in coming close to the ground-truth. This finding is consistent with other literature that has looked critically at SHAP and other xAI tools.

While posthoc-xAI does not make blackboxes interpretable, interpretability is not always unnecessary. Quantifying sample average treatment effects and making predictions does not require interpretable modeling [68], or even necessarily models at all [69]. Blackbox methods offer good performance with minimal thoughtfulness. For these reasons, blackbox methods remain inherently useful—so long as one does not whitewash them with false explainability.

### Critical look at the applied literature

As evidenced by the explosion of applied machine learning manuscripts that claim explainability or interpretability, the research community has realized that it is important to have some understanding of what a machine learning model is doing. In many cases, the claim of interpretability is warranted, for instance in manuscripts that use methods such as logistic regression or even more-advanced methods such as explainable-boosting machines [70, 71]. However, a large volume of studies (a small sampling for example [72–76]) are in-actuality putting forward blackboxes as *explainable* by using SHAP or similar methodologies.

Additionally, many manuscripts eschew the word explainable altogether and claim that their SHAP-endowed blackboxes are *interpretable*. It is our humble opinion that the machine learning field needs to set a higher bar for what should be labeled explainable, let along interpretable. It is our hope that we have made a contribution to this particular conversation.

## Limitations

Our modeling approach has downsides. Numerical stability when performing ADVI inference generally requires the use of double precision floating point—a limitation common to Bayesian inference (popular statistical packages such as Stan use double precision by default). This limitation is significant when looking to expand to larger models that encompass more data features. The methodology is based on piecewise linear modeling defined over a high-dimensional lattice of coarsened variables derived from the data. For this reason, models can become big quickly and can run into practical memory constraints, particularly in unison with the requirement for using double precision floating point.

The trend in machine learning has been to move towards more automation and the search for modeling architectures that do not require tuning beyond engineering of input predictors. Bucking this trend, our framework is designed with more intentionality in mind—it requires the practitioner to think about what types of broad interactions make sense for a given problem and what types of coarsening will give rise to a model that is understandable and useful in a real-world sense. Some may view this characteristic as a limitation.

## Supporting information

**S1 File. Supplementary methods and results.**
(PDF)

**S2 File. Supplementary review history from prior KDD submission.**
(PDF)

## Acknowledgments

We thank the Innovation Center of the Center for Medicare and Medicaid services for providing access to the CMS Limited Dataset through DUA LDSS-2019-54177. We also thank Dr. Pei-Shu Ho for help in understanding Medicare billing data.

## Author Contributions

**Conceptualization:** Ted L. Chang, Hongjing Xia, Sonya Mahajan, Rohit Mahajan, Joe Maisog, Shashaank Vattikuti, Joshua C. Chang.

**Data curation:** Joshua C. Chang.

**Formal analysis:** Ted L. Chang, Hongjing Xia, Joe Maisog, Shashaank Vattikuti, Joshua C. Chang.

**Funding acquisition:** Joshua C. Chang.

**Investigation:** Hongjing Xia, Joe Maisog, Shashaank Vattikuti, Carson C. Chow, Joshua C. Chang.

**Methodology:** Hongjing Xia, Sonya Mahajan, Rohit Mahajan, Shashaank Vattikuti, Carson C. Chow, Joshua C. Chang.

**Project administration:** Ted L. Chang, Hongjing Xia, Shashaank Vattikuti, Joshua C. Chang.

**Software:** Ted L. Chang, Hongjing Xia, Joshua C. Chang.

**Supervision:** Hongjing Xia, Sonya Mahajan, Carson C. Chow, Joshua C. Chang.

**Validation:** Ted L. Chang, Hongjing Xia, Joshua C. Chang.

**Visualization:** Ted L. Chang, Joshua C. Chang.

**Writing – original draft:** Sonya Mahajan, Rohit Mahajan, Shashaank Vattikuti, Joshua C. Chang.

**Writing – review & editing:** Ted L. Chang, Hongjing Xia, Carson C. Chow, Joshua C. Chang.

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
