## [Decision Letter · Decision Letter 0]

10 Jan 2024

PONE-D-23-32447Interpretable (not just posthoc-explainable) medical claims modeling for discharge placement to reduce preventable all-cause readmissions or deathPLOS ONE

Dear Dr. Chang,

Thank you for submitting your manuscript to PLOS ONE. After careful consideration, we feel that it has merit but does not fully meet PLOS ONE’s publication criteria as it currently stands. Therefore, we invite you to submit a revised version of the manuscript that addresses the points raised during the review process.

Please revise.

We look forward to receiving your revised manuscript.

Kind regards,

Academic Editor

PLOS ONE

“We thank the Innovation Center of the Center for Medicare and Medicaid services for

providing access to the CMS Limited Dataset through DUA LDSS-2019-54177. We also

thank Dr. Pei-Shu Ho for help in understanding Medicare billing data. CCC is

supported by the Intramural Research Program of the NIH, NIDDK. JCC is partially

supported by the Intramural Research Program of the NIH, Clinical Center. This work

used the Extreme Science and Engineering Discovery Environment (XSEDE) [58], which

is supported by National Science Foundation grant number ACI-1548562 through

allocation TG-DMS190042.”

“CCC is supported by the Intramural Research Program of the NIH, NIDDK. JCC is partially supported by the Intramural Research Program of the NIH, Clinical Center. This work used the Extreme Science and Engineering Discovery Environment (XSEDE) [58], which is supported by National Science Foundation grant number ACI-1548562 through allocation TG-DMS190042.”

5. In the online submission form, you indicated that your data is available only on request from a third party. Please note that your Data Availability Statement is currently missing the contact details for the third party, such as an email address or a link to where data requests can be made. Please update your statement with the missing information.

7. Please include a separate caption for each figure in your manuscript.

Reviewers' comments:

Reviewer's Responses to Questions

**Comments to the Author**

1. Is the manuscript technically sound, and do the data support the conclusions?

Reviewer #1: Yes

Reviewer #2: Yes

2. Has the statistical analysis been performed appropriately and rigorously? 

Reviewer #1: I Don't Know

Reviewer #2: Yes

3. Have the authors made all data underlying the findings in their manuscript fully available?

Reviewer #1: Yes

Reviewer #2: Yes

4. Is the manuscript presented in an intelligible fashion and written in standard English?

Reviewer #1: Yes

Reviewer #2: Yes

5. Review Comments to the Author

Reviewer #1: These investigators tested whether a Bayesian model would produce an equivalent outcomes prediction accuracy to deep neural networks but have the benefit of interpretability.

The statistical analytic approach exceeds my expertise as a clinical researcher and I would recommend a review by a data scientist or statistician who understands both Bayesian statistics and deep learning models. Consequently, I can only provide some genearl comments

1. The use of CCS codes is ideal for a model like this due to the granularity of the codes. It's nice to see a group expand beyond Charlson or Elixhauser -- which would limit an analysis like this.

2. The Introduction section seems long for a clinical paper -- but I'm unfamiliar with the structure of statistical papers and this could be typical

3. The Discussion does not compare and contrast with the existing literature. It would be helpful for the reader to comprehensively understand how the investigators' model compares to existing models on a more detailed level.

4. The Limitations section should be expanded and more specific.

Reviewer #2: The manuscript presents an interpretable multilevel Bayesian framework that captures the piecewise linearity of ReLU-activated deep neural networks. The framework is applied to develop a survival model for predicting hospital readmission and death using medical claims data, with a focus on discharge placement and adjusting for confounding variables. Trained on a 5% sample of Medicare beneficiaries, the model is tested on 2012 episodes and performs competitively with XGBoost and a Bayesian deep neural network, achieving an AUROC of approximately 0.75. The model emphasizes interpretability without sacrificing accuracy, providing a global interpretation of its reasoning and identifying relative risk factors while quantifying the impact of discharge placement. The study also critiques the posthoc explainer SHAP, suggesting inconsistencies with the ground truth model reasoning.

The model is novel and the literature in it serve as a comprehensive to the XAI models in the field. The paper is interesting and may gain so many interesting. However, I have some minor suggestions/concerns:

- Since the AUCROC is the main meaure of performance. I was hoping to see the AUCROC curvers for running the proposed model and the other compared models for multiple runinng points (TPRs, and FDRs).

- The abstract may add qauntitiy of the samples (around 1.2 million episodes) with/instead of the 5% of the data.

- For the scope of health care, I suggest to highlight more applications of XAI in health care. I suggest to highlight PMID: 38066735 and/or PMID: 38132885

- Please order the references [1],[2], ascendingly based on the appearance in the manuscript.

6. PLOS authors have the option to publish the peer review history of their article (what does this mean?). If published, this will include your full peer review and any attached files.

Reviewer #1: No

Reviewer #2: **Yes: **Abedalrhman Alkhateeb

---

## [Decision Letter · Decision Letter 1]

16 Apr 2024

Interpretable (not just posthoc-explainable) medical claims modeling for discharge placement to reduce preventable all-cause readmissions or death

PONE-D-23-32447R1

Dear Dr. Chang,

We’re pleased to inform you that your manuscript has been judged scientifically suitable for publication and will be formally accepted for publication once it meets all outstanding technical requirements.

Kind regards,

Academic Editor

PLOS ONE

Additional Editor Comments (optional):

Reviewers' comments:

Reviewer's Responses to Questions

**Comments to the Author**

1. If the authors have adequately addressed your comments raised in a previous round of review and you feel that this manuscript is now acceptable for publication, you may indicate that here to bypass the “Comments to the Author” section, enter your conflict of interest statement in the “Confidential to Editor” section, and submit your "Accept" recommendation.

Reviewer #1: All comments have been addressed

Reviewer #3: All comments have been addressed

2. Is the manuscript technically sound, and do the data support the conclusions?

Reviewer #1: Yes

Reviewer #3: Yes

3. Has the statistical analysis been performed appropriately and rigorously? 

Reviewer #1: Yes

Reviewer #3: Yes

4. Have the authors made all data underlying the findings in their manuscript fully available?

Reviewer #1: Yes

Reviewer #3: No

5. Is the manuscript presented in an intelligible fashion and written in standard English?

Reviewer #1: Yes

Reviewer #3: Yes

6. Review Comments to the Author

Reviewer #1: (No Response)

Reviewer #3: The manuscript adress an increasingly controversial issue, as is the interpretability os AI models predictions in Medicine. The methodology is sound and well described, though the readability of some of the methods and figures presented is hard to health professsionals with no expertise in high level statistics anbd computing.

7. PLOS authors have the option to publish the peer review history of their article (what does this mean?). If published, this will include your full peer review and any attached files.

Reviewer #1: No

Reviewer #3: No

---

## [Editor Report · Acceptance letter]

26 Apr 2024

PONE-D-23-32447R1 

PLOS ONE

Dear Dr. Chang, 

I'm pleased to inform you that your manuscript has been deemed suitable for publication in PLOS ONE. Congratulations! Your manuscript is now being handed over to our production team.

Kind regards, 

on behalf of

Dr. Robert Jeenchen Chen 

Academic Editor

PLOS ONE